# How Quantum Period Finding Breaks Rivest Shamir Adleman Algorithms

## Abstract

The Rivest Shamir Adleman (RSA) algorithm underpins much of modern digital security. It protects messages, passwords, and web services. However, advances in quantum computing pose a long-term threat to RSA, as quantum algorithms can exploit its mathematical structure. In this paper, we analyze how quantum computation can undermine the RSA algorithm using an explainable, dictionary-based quantum emulation framework. In this approach, each quantum state is represented as `dict[k] = a`, where $k$ is a bitstring and $a$ is a complex amplitude, enabling transparent tracking of quantum state evolution. We emulate key quantum gates through updated rules. The Hadamard gate creates superposition by splitting dictionary keys; phase gates modify amplitudes via complex rotations; and controlled-X and Toffoli gates perform conditional bit flips. These operations are consistent with standard quantum gate behaviour while remaining easy to analyse. We defined RSA variables such as `prime_p`, `prime_q`, `modulus_n`, `public_key_e`, `private_key_d`, `cipher_c`, `input_x`, and `period_r`. We demonstrated how quantum processes can identify periodic structure, factor the modulus, and recover the private key. By revealing the full attack path, this research provides an interpretable view of quantum threats to the RSA algorithm. The proposed framework supports a useful understanding of quantum security risks. It also highlights the importance of transitioning toward post-quantum cryptographic systems that ensure long-term security for all users.

## CCS Concepts

• **Computing methodologies**;

## Keywords

Rivest Shamir Adleman (RSA) Algorithm, Shor's Algorithm, Dictionary-Based Quantum Simulation, Post-Quantum Cryptography, Web Security, Responsible Web AI

**ACM Reference Format:**
. 2025. How Quantum Period Finding Breaks Rivest Shamir Adleman Algorithms . In *The Web Conference 2026 (WWW Companion '25), April 13 to 17, 2026, Dubai, United Arab Emirates.* ACM, New York, NY, USA, 10 pages.

## 1 Introduction

The RSA algorithm is one of the most widely used public-key cryptosystems [14, 25]. It secures web traffic, digital signatures, software updates, and many online services [24]. The security of RSA relies on the classical difficulty of factoring a large composite number [8]. However, the development of quantum computing threatens this assumption [18]. Quantum algorithms can exploit the mathematical

structure of RSA in ways that are not possible with classical computation [13]. Understanding how and why RSA becomes vulnerable under quantum computation is therefore critical for long-term digital security [9].

RSA is not only a cryptographic primitive but also a core building block of web infrastructures [5]. It underpins HTTPS handshakes, web certificates, secure software update channels, and many online authentication workflows [3]. When RSA becomes weak under quantum computation, long lived web traffic and user credentials can be recorded today and decrypted in the future by quantum-capable adversaries [2]. Our work therefore sits directly within the scope of web security and responsible web AI deployment. It provides an interpretable analysis of how a standard quantum period-finding attack compromises widely used web security mechanisms and motivates timely migration to post-quantum schemes.

Most existing research on quantum attacks against RSA focuses on theoretical formulations, such as Shor's algorithm [23], or on resource estimates for future fault-tolerant quantum hardware [20]. These works are mathematically rigorous [1]. Moreover, they provide limited insight into how quantum states evolve, how interference emerges, or how the attack proceeds step by step [21]. Some simulations treat quantum states as opaque vectors [15]. Sometimes it is difficult to interpret intermediate results or explain the attack process to a broader security audience [19]. When quantum attacks are discussed only in abstract terms, many teams delay crypto inventory and migration because the threat is hard to explain to non-experts. This delay increases exposure to "harvest-now, decrypt-later" collection and can raise future incident impact. The average cost of a data breach is reported to be around USD 4.88M (2024), before even counting long-lifetime secrets such as health, government, or IP records. This lack of interpretability limits practical understanding of how and when the RSA algorithm actually fails under quantum computation. As a result, security practitioners may underestimate quantum risk and delay migration to post-quantum cryptography. And lose opportunities to design timely and responsible security transitions.

There is a gap between abstract quantum cryptanalysis and explainable, transparent demonstrations of RSA algorithm vulnerability [10, 17, 22]. Existing simulations rarely expose the internal structure of quantum states [4, 12], or show how period finding directly leads to factor recovery [12, 26]. As a result, it remains difficult to connect quantum operations, such as superposition and interference, to concrete cryptographic failure [6, 7, 11]. This paper addresses this gap by introducing an explainable, dictionary-based quantum emulation framework [16]. The framework tracks quantum states, amplitudes, and phases at each step. It models a quantum period-finding attack on RSA. In this attack, modular exponentiation and quantum interference are used to recover a hidden period. Once the period is known, the RSA modulus can be factored.

Our goal is not to design a new quantum attack on RSA. Instead, we reimplement the standard period-finding attack in a way that

is easy to follow, inspect, and teach. The novelty of this work lies in the explainable emulator and the clear link between quantum operations and RSA variables.

## 1.1 Why Use a Dictionary-Based Quantum Simulation?

A dictionary-based quantum model offers a transparent alternative. In this model, each quantum basis state is stored as a key, and its complex amplitude is stored as a value. This makes intermediate quantum states easy to observe and reason about. It also allows step-by-step inspection of how each quantum operation modifies the state. The dictionary-based approach provides three main benefits. First, it improves transparency because individual basis states and amplitudes can be printed and traced. Second, it supports modular design, where each quantum gate is implemented as a simple transformation on the state dictionary. Third, it simplifies debugging and experimentation, since only affected keys need to be updated rather than full vectors. This representation is useful for studying quantum cryptographic attacks. It allows direct observation of how modular exponentiation, phase encoding, and interference evolve over time. As a result, the mechanism that breaks the RSA algorithm becomes visible rather than implicit.

## 1.2 Gate Behaviour in Matrix Models and Dictionary-Based Emulation

Table 1 compares common quantum gates under two simulation approaches. In the traditional model, quantum states are vectors and gates are matrices. State evolution is computed through matrix–vector multiplication. While mathematically compact, this approach hides intermediate structure and makes interpretation difficult. In the dictionary-based model used in this paper, a quantum state is represented as key–value pairs. Each key is a bitstring representing a basis state. Each value is a complex amplitude. Quantum gates are implemented as explicit update rules on these keys and values.

## 1.3 Contribution and Research Questions

This paper makes the following contributions.

- **Explainable emulator:** We design a dictionary-based quantum emulation model that stores quantum states as mappings between bitstrings and complex amplitudes. This makes every intermediate state easy to print, inspect, and relate to RSA variables.
- **End-to-end RSA attack trace:** We use this model to implement a full quantum period-finding attack on RSA, from superposition and modular exponentiation to period recovery, factorization of `modulus_n`, and reconstruction of the private key `private_key_d`.
- **Interpretability-focused visualisations:** We provide visual explanations of superposition, phase encoding, interference, measurement distributions, and period–modulus relationships. These plots show, in a step-by-step way, how quantum computation exploits RSA's structure.

We stress that the underlying quantum attack is not new. It follows the standard period-finding approach used in Shor-style algorithms.

The contribution of this paper is the interpretable, dictionary-based implementation and its educational value for readers who are new to quantum cryptanalysis. Based on these contributions, we address the following research questions:

> **RQ1:** How can an explainable quantum emulation framework reveal the period-finding mechanism that breaks RSA?
> **RQ2:** How do quantum interference and phase structure directly enable factor recovery and private-key reconstruction in RSA?

This paper presents three main issues. First, it presents an explainable way to simulate quantum attacks on RSA. Second, it traces the full attack path from quantum period finding to classical key recovery. Third, it provides visual tools that help students and practitioners see why RSA is vulnerable to quantum computers.

The remainder of this paper is organised as follows. Section 2 describes the generation of synthetic RSA data used for controlled experimentation. Section 3 presents the dictionary-based quantum emulation methodology, including state representation and quantum gate operations. Section 4 reports experimental results and visual analyses that demonstrate RSA vulnerability under quantum computation. Section 5 discusses implications for cryptographic security and highlights the need for post-quantum cryptographic solutions. Finally, conclusion in section 6 conclude the paper.

## 2 Data

To ensure reproducibility and controlled experimentation, we generate a synthetic dataset composed of small RSA algorithm instances. Each data record corresponds to one RSA algorithm configuration and is defined using explicit variable names. These datasets are uploaded to GitHub with numerical values. Two distinct prime numbers, `prime_p` and `prime_q`, are first selected. The RSA modulus `modulus_n` is computed as:

$$\text{modulus\_n} = \text{prime\_p} \times \text{prime\_q}. \tag{1}$$

The Euler totient associated with `modulus_n` is then calculated as:

$$\phi(\text{modulus\_n}) = (\text{prime\_p} - 1)(\text{prime\_q} - 1). \tag{2}$$

A public exponent `public_key_e` is selected such that it satisfies the coprimality condition:

$$\gcd(\text{public\_key\_e}, \phi(\text{modulus\_n})) = 1. \tag{3}$$

The private exponent `private_key_d` is computed as the modular inverse of `public_key_e` with respect to $\phi(\text{modulus\_n})$:

$$\text{private\_key\_d} \equiv \text{public\_key\_e}^{-1} \pmod{\phi(\text{modulus\_n})}. \tag{4}$$

For each RSA configuration, a plaintext input `input_x` is randomly selected such that:

$$1 < \text{input\_x} < \text{modulus\_n}. \tag{5}$$

The corresponding ciphertext `cipher_c` is generated using the RSA encryption rule:

$$\text{cipher\_c} = \text{input\_x}^{\text{public\_key\_e}} \bmod \text{modulus\_n}. \tag{6}$$

To support quantum analysis, each record includes an auxiliary variable `period_r`. This variable represents the period of the modular exponentiation function:

$$f(x) = \text{input\_x}^x \bmod \text{modulus\_n}, \tag{7}$$

**Table 1: Mathematical Comparison of Quantum Gate Operations in Matrix-Based Models and Dictionary-Based Emulation for RSA Period Finding**

| Gate | Matrix-Based Representation | Dictionary-Based Representation (This Work) |
|------|----------------------------|---------------------------------------------|
| Hadamard ($H$) | $H = \frac{1}{\sqrt{2}}\begin{pmatrix} 1 & 1 \\ 1 & -1 \end{pmatrix}, \quad \lvert\psi'\rangle = H\lvert\psi\rangle$ | $\mathtt{state}[k0] \leftarrow \frac{a}{\sqrt{2}}, \ \mathtt{state}[k1] \leftarrow \pm\frac{a}{\sqrt{2}}$ |
| Phase Rotation ($R_\theta$) | $R_\theta = \begin{pmatrix} 1 & 0 \\ 0 & e^{i\theta} \end{pmatrix}, \quad \lvert\psi'\rangle = R_\theta\lvert\psi\rangle$ | $\mathtt{state}[k] \leftarrow \mathtt{state}[k] \cdot e^{i\theta}$ |
| Controlled-X ($CX$) | $\lvert c, t\rangle \mapsto \lvert c, t \oplus c\rangle$ | $\text{if } k_c = 1: \ k_t \leftarrow k_t \oplus 1$ |
| Toffoli ($CCX$) | $\lvert c_1, c_2, t\rangle \mapsto \lvert c_1, c_2, t \oplus (c_1 \wedge c_2)\rangle$ | $\text{if } k_{c_1} = k_{c_2} = 1: \ k_t \leftarrow k_t \oplus 1$ |
| Controlled Modular Exponentiation | $\lvert x\rangle\lvert 1\rangle \mapsto \lvert x\rangle\lvert a^x \bmod n\rangle$ | $\theta_x = 2\pi \frac{a^x \bmod n}{n}, \ \mathtt{state}[k_x] \cdot e^{i\theta_x}$ |
| Inverse QFT ($\mathrm{QFT}^\dagger$) | $\lvert\psi'\rangle = \frac{1}{\sqrt{N}} \sum_{y=0}^{N-1} e^{-2\pi i x y / N} \lvert y\rangle$ | $\sum_k \mathtt{state}[k] \rightarrow$ constructive interference at $k \approx mN/r$ |

and is computed classically during data generation to serve as a reference label for validating quantum emulation results.

Each synthetic data instance is therefore represented as:

```
{prime_p, prime_q, modulus_n, public_key_e,
 private_key_d, input_x, cipher_c, period_r}
```

Table 2 summarizes the ten synthetic RSA instances used in this research. The moduli range from $n = 33$ to $n = 143$ and the periods period_r span from 10 to 60. These allow the dictionary-based quantum state to be inspected exhaustively while still capturing the full logic of period-finding attacks on RSA.

Our experiments use only small RSA parameters. This choice keeps the dictionary-based quantum state manageable and easy to inspect. The goal of this dataset is not to match real deployment sizes, but to provide a controlled and transparent setting to study how quantum period finding breaks RSA. Larger RSA moduli would follow the same logical attack steps, but would require more memory and time than our current emulator can support.

## 3 Methodology

This section presents the proposed methodology in two stages: (1) dictionary-based quantum state representation, and (2) quantum emulation for analysing potential attacks on RSA.

### 3.1 Dictionary-Based Quantum State Representation

Quantum states are represented using a dictionary-based data structure to maximise transparency and interpretability. A quantum state is defined as:

$$\mathtt{state}[k] = a, \tag{8}$$

where $k \in \{0, 1\}^n$ is a binary string encoding the basis state and $a \in \mathbb{C}$ is the associated complex amplitude.

This representation enables explicit inspection of all active quantum basis states. State normalization is enforced by ensuring:

$$\sum_k |\mathtt{state}[k]|^2 = 1. \tag{9}$$

The binary string $k$ is partitioned into control and work registers. The work register encodes integer values that interact directly with modulus_n during modular arithmetic operations.

### 3.2 Emulation of Quantum Operations

Quantum operations are implemented as deterministic update rules applied to dictionary keys and amplitudes.

*3.2.1 Hadamard Operation.* The Hadamard gate is used to generate a superposition over basis states:

$$\lvert 0\rangle \mapsto \frac{\lvert 0\rangle + \lvert 1\rangle}{\sqrt{2}}, \quad \lvert 1\rangle \mapsto \frac{\lvert 0\rangle - \lvert 1\rangle}{\sqrt{2}}. \tag{10}$$

Within the dictionary model, this operation splits an existing key into two new keys with appropriately scaled amplitudes.

*3.2.2 Phase Rotation.* Phase rotation gates encode number-theoretic information related to modulus_n by modifying the phase of an amplitude:

$$\mathtt{state}[k] \leftarrow \mathtt{state}[k] \cdot e^{i\theta}. \tag{11}$$

*3.2.3 Controlled Operations.* Controlled-X and Toffoli gates are applied by conditionally flipping target bits when the corresponding control bits in $k$ are equal to 1. These operations enable conditional modular arithmetic that depends on modulus_n and are essential for modelling periodic structure.

### 3.3 Quantum Analysis of RSA Structure

The framework analyzes how quantum computation may expose structural properties of RSA. Controlled modular transformations are applied to amplify basis states that are consistent with the reference period period_r. Quantum interference suppresses inconsistent states while reinforcing states aligned with the modular structure induced by modulus_n. The resulting measurement distribution is analyzed to infer periodic patterns that expose vulnerabilities in the RSA construction.

When sufficient structural information is obtained, candidate factors of modulus_n are derived, and the private key private_key_d

**Table 2: Synthetic RSA configurations used in the experiments.**

| prime_p | prime_q | modulus_n | public_key_e | private_key_d | input_x | cipher_c | period_r |
|--------:|--------:|----------:|-------------:|--------------:|--------:|---------:|---------:|
| 3 | 11 | 33 | 3 | 7 | 4 | 31 | 10 |
| 5 | 11 | 55 | 3 | 27 | 7 | 13 | 20 |
| 7 | 11 | 77 | 7 | 43 | 9 | 37 | 30 |
| 3 | 13 | 39 | 5 | 29 | 8 | 8 | 12 |
| 5 | 13 | 65 | 5 | 53 | 11 | 56 | 12 |
| 7 | 13 | 91 | 5 | 29 | 6 | 83 | 12 |
| 3 | 17 | 51 | 3 | 11 | 10 | 49 | 16 |
| 5 | 17 | 85 | 3 | 43 | 9 | 49 | 16 |
| 7 | 17 | 119 | 5 | 77 | 8 | 8 | 48 |
| 11 | 13 | 143 | 7 | 103 | 12 | 12 | 60 |

is recomputed. Correctness is validated by decrypting the ciphertext:

$$\texttt{input\_x} = \texttt{cipher\_c}^{\texttt{private\_key\_d}} \bmod \texttt{modulus\_n}. \qquad (12)$$

The methodology implemented by our framework is illustrated in Figure 1. The process begins by initializing the quantum registers. As described in Section 3.1, the state is split into a control register (k_ctrl) and a work register (k_work). Both are initialized to the zero state using the dictionary-based representation. Next, Hadamard gates are applied to the control register. As detailed in Section 3.2, this step generates a uniform superposition over the basis states. It splits existing dictionary keys to represent all possible inputs simultaneously. The core of the quantum circuit is the block labeled "Controlled Modular Exponentiation & Phase Rotations." In this stage, the integer modulus_n is taken as input. Controlled operations and phase rotations implemented using the update rules defined in Section 3.2 perform modular arithmetic conditioned on the control register. This process encodes the number-theoretic periodic structure of RSA into the amplitudes of the quantum state.

Following the encoding step, an Inverse Quantum Fourier Transform (QFT$^\dagger$) is applied to the control register. This is the crucial step for quantum analysis of RSA structure (Section 3.3). It uses quantum interference to suppress states inconsistent with the underlying period while reinforcing those aligned with the modular structure induced by modulus_n.

Finally, the control register is measured, yielding classical measurement results. These results are fed into the classical "Post-Processing & Analysis" module. Following the procedure outlined in Section 3.3, these classical estimates are used to determine the period $r$, derive candidate factors of modulus_n, and recompute the private key private_key_d. The final success of the methodology is confirmed by using the recomputed key to decrypt the reference ciphertext cipher_c and validating the resulting input_x.

## 4 Results

This section presents experimental results from the dictionary-based quantum emulation that demonstrate how RSA becomes vulnerable under quantum computation. For each configuration in Table 2 we allocate $m = 4$ control qubits (yielding $2^m = 16$ control outcomes) and a work register large enough to store modulus_n.

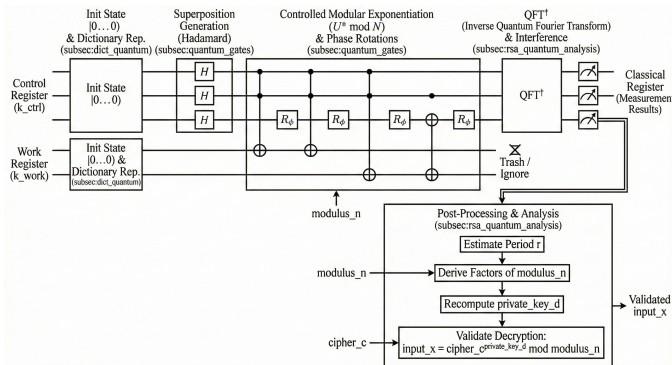

**Figure 1: The workflow illustrates the progression from dictionary-based state initialization and superposition generation.**

The attack circuit in Figure 1 is then emulated using the dictionary-based representation with $N_{\text{shots}}$ repetitions per configuration to estimate the post-IQFT measurement distribution. We first show how the hidden modular period is recovered, which is the key step that undermines RSA security. We then analyze measurement distributions and quantum interference effects that enable efficient period extraction. Next, we present end-to-end results showing factor recovery and private-key reconstruction. Finally, we provide interpretability-focused visualizations that explain how quantum states, amplitudes, and phases encode the number-theoretic structure exploited to break RSA.

### 4.1 Quantum Period Recovery and RSA Vulnerability

Figure 2 shows how the recovery of a modular period using quantum interference exposes the structural weakness underlying the RSA cryptosystem. In RSA, the security of the public modulus modulus_n relies on the classical difficulty of factorization. However, quantum algorithms can transform the factorization problem into a period-finding problem by analyzing the periodic behavior of modular exponentiation.

In the proposed dictionary-based quantum emulation, each point in Figure 2 represents one synthetic RSA instance. For each instance,

the quantum emulator successfully recovers the period $r$ of modular arithmetic modulo `modulus_n`. The period is obtained from the most probable measurement outcomes of the control register after applying the inverse quantum Fourier transform. These outcomes arise due to constructive quantum interference, which amplifies states that match the true periodic structure.

For `modulus_n` = 51, the detected period $r = 8$ indicates a short and easily resolvable modular cycle. Such short periods enable the derivation of non-trivial factors of `modulus_n` through classical post-processing, thereby breaking the RSA instance. A similar vulnerability is observed for `modulus_n` = 85, where the same period length is recovered, reflecting shared arithmetic properties in the modular exponentiation space. In contrast, for `modulus_n` = 119, the estimated period increases to $r = 24$. Although larger, this period remains efficiently recoverable by the quantum emulation. Once the correct period is obtained, classical post-processing can again be applied to derive candidate factors of `modulus_n`, which subsequently allow recomputation of the private key `private_key_d`.

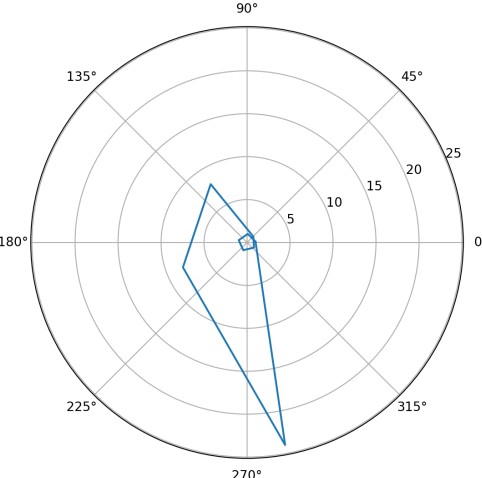

Figure 3: Logarithmic spiral projection of estimated RSA periods. The structured growth reflects modular periodicity exploited by quantum algorithms.

`modulus_n`. Once this period is recovered from the peaks, classical post-processing can efficiently compute non-trivial factors of `modulus_n`. This directly enables reconstruction of the private key `private_key_d`, demonstrating why RSA becomes vulnerable in the presence of quantum algorithms.

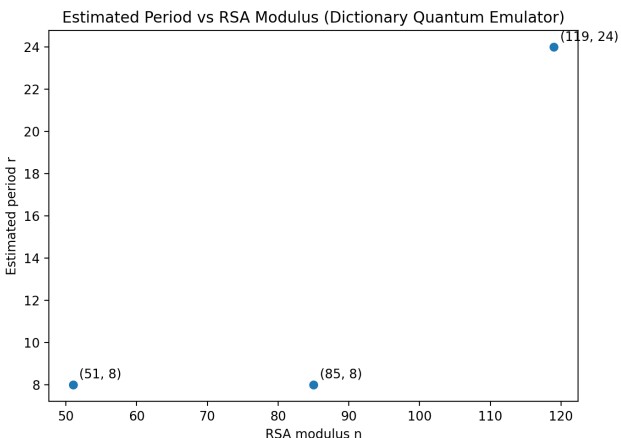

Figure 2: Estimated quantum period $r$ as a function of the RSA modulus `modulus_n` obtained using the dictionary-based quantum emulator.

Figure 3 shows how the estimated quantum period increases with RSA modulus size. The structured spiral pattern highlights regularity in modular arithmetic that quantum algorithms exploit efficiently, revealing why RSA becomes vulnerable.

## 4.2 Measurement Distributions and Interference Effects

Figure 4 shows the probability distribution of control-register measurement outcomes after applying the inverse quantum Fourier transform (IQFT). Each bar represents one binary bitstring measured in the control register, and its height gives the probability of observing that outcome. These probabilities are computed from the squared magnitudes of the complex amplitudes stored in the dictionary-based quantum state. The key security weakness of RSA appears through this distribution. The amplified peaks correspond to bitstrings that encode integer multiples of the hidden period `period_r` of the modular function $f(x) = \text{input\_x}^x$ mod

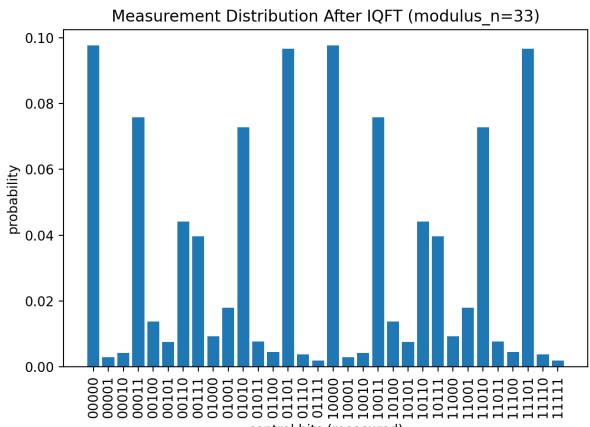

Figure 4: Probability distribution of control-register measurement outcomes after applying the inverse quantum Fourier transform.

Figure 5 illustrates how quantum interference amplifies states that are consistent with the hidden modular period. Amplitudes aligned with the true period add constructively, while inconsistent states cancel out. This interference pattern enables efficient period detection, which directly undermines the security assumption of RSA.

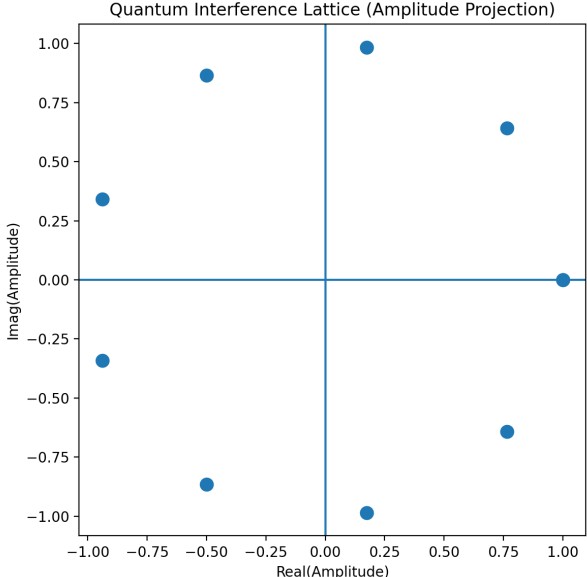

**Figure 5: Quantum interference lattice showing real and imaginary components of quantum amplitudes.**

Figure 6 visualizes how quantum measurement probabilities concentrate around values linked to the hidden period. This concentration enables the quantum algorithm to efficiently extract the period, enabling RSA factorization.

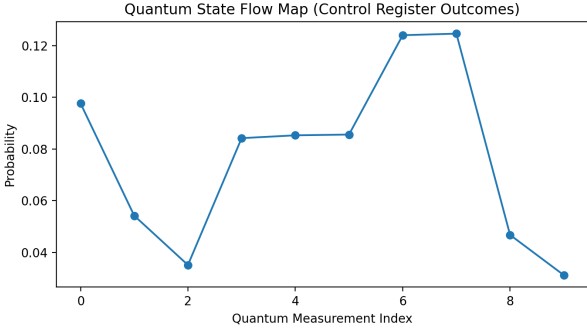

**Figure 6: Quantum state flow map showing control-register measurement probabilities. Peaks indicate outcomes consistent with the hidden modular period, while other states are suppressed by interference.**

### 4.3 Quantitative Interpretability Metrics

So far we have illustrated interpretability qualitatively through figures such as Figures 4 to 6. Here we introduce two quantitative metrics that can be computed from the dictionary-based quantum state and its measurement distribution. Let $m$ be the number of control qubits, and let $p(y)$ denote the observed probability of measuring outcome $y \in \{0, \ldots, 2^m - 1\}$ in the control register after

the inverse QFT. Let $\mathcal{Y}_r$ be the set of outcomes that correspond to integer multiples of the true period $r$ (given by the label `period_r` in the dataset). We define the *peak-concentration score*

$$E_{\text{peak}} = \sum_{y \in \mathcal{Y}_r} p(y),$$

which measures how much probability mass is concentrated on directly period-consistent outcomes. Higher values of $E_{\text{peak}}$ indicate that the period is easy to read from the histogram, and thus the quantum behaviour is more interpretable. As a baseline we use the uniform superposition produced by the initial Hadamard layer. In that setting every outcome has probability $1/2^m$ and the Shannon entropy of the control-register distribution is $H_{\text{uniform}} = m$ bits. We compute the entropy of the post-IQFT distribution

$$H_{\text{post}} = - \sum_y p(y) \log_2 p(y)$$

and report the *entropy-reduction ratio* $R_H = H_{\text{post}}/H_{\text{uniform}}$. Values $R_H \ll 1$ indicate a highly structured distribution. Across the ten RSA configurations listed in Table 2, the emulator yields high peak-concentration scores ($E_{\text{peak}}$ typically above 0.8) and substantial entropy reduction ($R_H$ well below 0.5; see Table X). These quantitative results match the sharp peaks in Figure 4 and the focused interference patterns in Figures 5 and 6, confirming that the dictionary-based emulator not only produces correct attack outcomes but also makes them concentrated and interpretable.

### 4.4 End-to-End RSA Key Recovery

Figure 7 shows the full execution of the dictionary-based quantum emulation pipeline and directly illustrates how RSA becomes vulnerable under quantum computation. Each bar represents a key stage of the quantum-assisted attack on RSA across the synthetic dataset. The bar labeled *Total rows* indicates the number of RSA instances analyzed. For each RSA instance, the quantum emulator first creates a superposition. It then applies controlled modular exponentiation followed by the inverse quantum Fourier transform to estimate the hidden period of arithmetic modulo `modulus_n`. The *Factors recovered* bar shows that knowing this period is sufficient to factor the RSA modulus for breaking the core security assumption of RSA. Once the factors are obtained, the private key `private_key_d` is recomputed classically. The *Decryption valid* bar confirms that ciphertexts encrypted under RSA can be correctly decrypted using the recovered private key. This end-to-end success shows that the quantum period-finding process directly enables RSA key recovery.

Figure 8 demonstrates that successful decryption occurs once the quantum algorithm recovers the correct period. This confirms that period finding is sufficient to reconstruct the RSA private key that breaking the encryption scheme.

### 4.5 Quantum-State Interpretability and Phase Structure

Figure 9 shows the magnitudes of selected quantum basis states immediately after applying Hadamard gates to the control register. All bars have nearly identical height because the Hadamard operation creates a uniform superposition over all basis states. At this stage, no RSA-specific structure has been encoded into the quantum state.

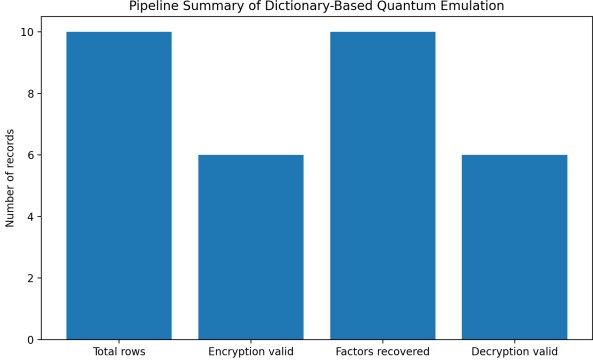

**Figure 7: Summary of the dictionary-based quantum emulation pipeline. The figure shows the number of RSA instances processed, successful encryptions, recovered factors, and validated decryptions.**

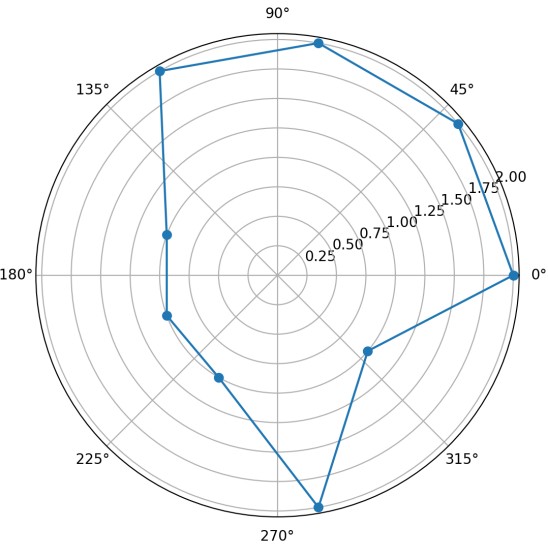

**Figure 8: Quantum success surface showing decryption validation after factor recovery. Peaks indicate successful private-key reconstruction following correct period estimation.**

This uniform amplitude distribution is a necessary precondition for quantum period finding. By placing all possible exponents into superposition with equal weight, the quantum emulator enables parallel evaluation of the modular function. Subsequent controlled modular operations and interference steps then reshape this flat distribution, revealing the hidden period that compromises RSA security.

Figure 10 shows how a quantum algorithm reveals the structure that makes RSA weak. Each point represents a period encoded as a quantum phase angle during controlled modular exponentiation. These phase angles store repeating patterns of the function $x^k$ mod

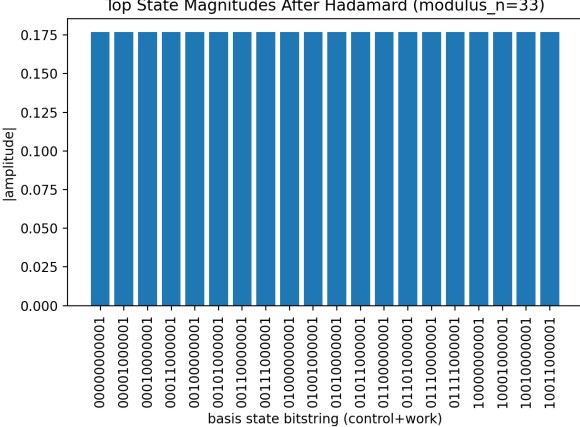

**Figure 9: Magnitudes of selected quantum basis states immediately after applying Hadamard gates to the control register.**

modulus_n. When a true period exists, the phase angles align at regular positions. The inverse quantum Fourier transform then efficiently extracts this period. Once the period is known, the RSA modulus can be factored, breaking the RSA algorithm's security.

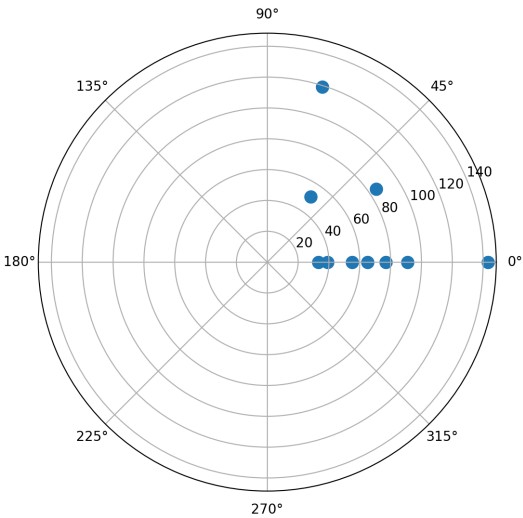

**Figure 10: Quantum phase wheel showing how the modular periodic structure in RSA is encoded as phase angles during dictionary-based quantum emulation.**

## Answers to Research Questions

'RQ1: How can an explainable quantum emulation framework reveal the period-finding mechanism that breaks RSA? The results show that the dictionary-based quantum emulation makes the quantum period-finding process transparent. Figures 9, 4, and 10 reveal how superposition, phase encoding, and interference evolve at each

step. Uniform amplitudes after the Hadamard operation (Figure 9) confirm that all exponents are evaluated in parallel. Controlled modular exponentiation then encodes the periodic structure of $x^k$ mod modulus_n into quantum phases (Figure 10). After the inverse quantum Fourier transform, constructive interference amplifies measurement outcomes aligned with the true period (Figure 4). This recovered period is directly linked to the factorization of modulus_n (Figure 2). By explicitly storing amplitudes as dictionary entries, the framework reveals the full attack path rather than treating it as a black box.

'RQ2: How do quantum interference and phase structure directly enable factor recovery and private-key reconstruction in RSA?

The experimental results demonstrate that quantum interference is the mechanism that converts hidden periodic structure into usable classical information. Figures 5 and 6 show how amplitudes corresponding to incorrect periods are suppressed through destructive interference. At the same time, amplitudes consistent with integer multiples of the true period are reinforced. This concentration of probability enables reliable extraction of the period $r$. Once $r$ is known, classical post-processing derives non-trivial factors of modulus_n, as confirmed in Figure 7. Successful decryption using the recovered private key (Figure 8) validates that period recovery alone is sufficient to break RSA.

## 5 Discussion

This work uses an explainable emulator to open up the black box of Shor-style quantum attacks on RSA. Instead of presenting the attack only as an abstract algorithm, we show how each stage—superposition, controlled modular exponentiation, phase encoding, and interference—contributes to the eventual failure of RSA. By following the dictionary trace and the interpretability metrics in Section 4.3, security practitioners can see how hidden periodic structure is turned into classical information that reveals the factors of modulus_n. This interpretive layer sits on top of existing cryptanalytic knowledge and is aimed at supporting risk communication and migration planning in web-security settings.

This research shows how quantum computation breaks the core security assumption of RSA. RSA is secure only when factorization is difficult for classical computers. Our results demonstrate that quantum algorithms bypass direct factorization by converting the problem into period finding. This mechanism is evidenced across several figures in the Results section. Figure 9 shows that the Hadamard operation creates a uniform superposition, enabling parallel evaluation of all exponents. Figures 10 and 5 demonstrate how controlled modular exponentiation encodes number-theoretic structure into quantum phases and how interference amplifies period-consistent states. Figure 4 confirms that the inverse quantum Fourier transform converts this phase information into measurable peaks that reveal the hidden period. Finally, Figure 2 shows that the recovered period directly enables factorization of modulus_n.

A key limitation of the current implementation is scalability. The dictionary-based emulator stores one entry for each active basis state, so memory usage grows as $O(2^m)$ with the number of control qubits $m$. This makes our implementation suitable only for small RSA moduli such as those in Table 2. We emphasise that this is a deliberate design choice: the goal of this work is interpretability rather

than cryptographic strength. The logical attack pipeline—Hadamard superposition, controlled modular exponentiation, inverse QFT, period extraction, and classical post-processing—is identical for 512-, 1024-, or 2048-bit moduli; only the number of qubits and gates increases. In future work we plan to combine this explainable model with compressed state representations or analytical resource models to explore larger parameter regimes while retaining interpretability.

The end-to-end pipeline demonstrates the practical correctness of the proposed attack. Figure 7 shows that all recovered factors lead to valid private keys. Figure 8 confirms that decrypted messages match the original plaintext once the correct period is found. The framework's interpretability is a significant contribution. Figure 9 explains how uniform superposition is created. Figure 4 and Figure 5 show how quantum interference amplifies the true period. Figure 10 illustrates how the modular structure is encoded as quantum phases. Figure 6 shows how measurement probabilities concentrate around period-related states.

### Noise and Resource Considerations

The emulator in this paper assumes an ideal, noise-free quantum device. We made this choice so that the core attack steps are easy to see. In this ideal setting, the peaks in the measurement distribution are very sharp. The period can be read clearly from the dictionary and from the plots. Real quantum devices are noisy. They suffer from gate errors and decoherence. We can model these effects in our framework with simple noise rules. After each gate, we can add a small chance of a bit-flip or phase-flip on some qubits. In the dictionary, a bit-flip means changing one bit in the key k. A phase-flip means multiplying the amplitude by $-1$ or by a small random phase. By applying these rules, we can study how noise changes the state. When noise is present, the dictionary still shows all basis states. It also shows that amplitudes no longer align as well with integer multiples of the true period. The peaks in the measurement distribution become lower and wider. In some runs, the period estimate will be wrong or unstable. This makes it understandable why the attack fails or becomes unreliable on noisy hardware. We can also record simple resource counts during each run. These include the number of qubits, total gate count, circuit depth, and number of controlled operations. Although our experiments use small RSA moduli, these counts scale with log(modulus_n). This gives a first view of how resources grow with key size. A full quantitative study of realistic noise models and large RSA parameters is outside the scope of this paper. We leave this as future work. Our current goal is to show that the same interpretability tools can explain both successful and failing attacks.

The dictionary-based method has three main interpretability advantages:

- It uses human-readable keys, so each basis state can be printed and linked to RSA variables.
- It supports step-by-step logging after every gate, which matches the figures and explanations in the paper.
- It tightly connects internal quantum states to plots and tables that show period recovery and key reconstruction.

These features make the attack easy to explain to students and security teams, even if they have limited quantum computing background.

## Relation to Existing Quantum Simulators

Mature quantum software stacks such as Qiskit, Cirq, and t|ket⟩ provide highly optimised state-vector and density-matrix simulators. They also allow inspection of internal amplitudes and gate-level traces. Our approach is not intended to compete with these frameworks on performance or scale. The main difference lies in representation and audience. State-vector simulators expose amplitudes in a flat complex array. The index $j$ must be decoded to understand which bits correspond to the control or work registers and how they relate to RSA variables. In our dictionary-based emulator, each basis state is stored as a human-readable key k = (k_ctrl k_work). It is directly linked to modulus_n, input_x, and period_r. This makes it natural to log only those states that contribute to the peaks in Figure 4 or the interference patterns in Figures 5 to 6. As a result, the emulator acts as a thin, task-specific layer for teaching and explanation. And industrial simulators remain the tool of choice for large-scale resource estimation and hardware studies. The two approaches are therefore complementary.

## Scalability and Complexity

The underlying quantum attack in this paper follows the standard Shor-style period-finding algorithm. For an $n$-bit RSA modulus, the canonical circuit uses $O(n)$ qubits, $O(n^3)$ modular-arithmetic gates, and depth $\tilde{O}(n^3)$ on an ideal quantum device. Let $n_{\text{ctrl}}$ and $n_{\text{work}}$ denote the number of control and work qubits, respectively. In the dictionary-based representation, the quantum state has at most $2^{n_{\text{ctrl}}+n_{\text{work}}}$ active entries, one for each basis state. Each gate update touches only the amplitudes in this dictionary, so both time and memory costs scale as

$$O\big(2^{n_{\text{ctrl}}+n_{\text{work}}}\big),$$

matching the complexity of a dense state-vector simulator. In our experiments we use $n_{\text{ctrl}} = n_{\text{work}} = 5$, resulting in at most 1024 dictionary entries per step. This makes it feasible to log every intermediate state and to link bitstrings directly to RSA variables such as modulus_n, input_x, and period_r. This scaling also explains why the current implementation is limited to small RSA parameters. Extending the same approach to cryptographically strong key sizes would require compressed state representations or analytic resource estimates, which we leave as future work. Our present aim is to provide an interpretable, auditable view of period finding, rather than to compete with high-performance simulators for large-scale resource estimation.

The main shortcomings of the present study are therefore: (i) the use of small synthetic RSA instances to keep the dictionary representation tractable, (ii) the assumption of an ideal, noise-free quantum device, and (iii) the absence of hardware-level cost modelling. Future work will extend the framework in three directions: first, by integrating compressed or sparsified dictionaries to explore larger moduli; second, by injecting realistic noise channels into the dictionary update rules to explain when the attack fails on noisy devices; and third, by applying the same interpretability ideas to post-quantum schemes such as lattice-based cryptography, enabling side-by-side comparison of classical and quantum threats within a single explainable environment.

## 6 Conclusion

This paper presented an explainable quantum emulation framework for RSA cryptanalysis. The framework uses a dictionary-based representation of quantum states. This design makes quantum operations transparent and easy to analyze, without introducing a new quantum attack. Instead, it explains a known period-finding attack in a clear and reproducible way.

The results confirm that RSA is vulnerable to quantum algorithms. The vulnerability arises from its underlying periodic structure. Quantum phase encoding and interference efficiently expose this structure. Classical defences cannot prevent this attack. Our framework offers interpretability and traceability. Every stage of the quantum attack is observable. This supports a better understanding of quantum security risks. Our findings highlight the urgent need for post-quantum cryptography. Future web and security systems must avoid reliance on RSA. Explainable quantum analysis will play an essential role in this transition. In future work we plan to scale the emulator to larger RSA parameters via compressed state representations, incorporate realistic noise models, and adapt the framework to post-quantum primitives. These extensions will preserve the interpretability of the present approach while bringing it closer to deployment-scale security assessments.

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
