# OpenReview forum: "How Quantum Period Finding Breaks Rivest Shamir Adleman Algorithms"
_ACM.org/TheWebConf/2026/Workshop/TIME — TIME 2026 Oral_

### Official Review · Reviewer_ijvC · 2025-12-30
**Outside Scope and Shallow**

**Rating:** 2
**Confidence:** 2

**Review:**

The RSA/quantum computing addresses cryptographic security and quantum algorithm implementation, which is OUTSIDE THE CONFERENCE'S SCOPE.

The paper explains the gap of explainability in RSA attacks in quantum. Then it proposes a novel approach to understanding quantum attacks: by representing quantum states as dictionaries to allow to explicitly track how quantum information evolves through computation (auditable). The paper is organized well. The paper ends by emphasizing the need of migrating away from RSA towards the post-quantum cryptography.

- No quantitative Metrics for "interpretability" (how this was validated, what is the baseline)
- No detail of experiment settings. Also modulus used (51,85,119) are smaller than industry standards (512, 1024, 2048)
- Complexity/cost analysis/comparison is missing (and what is the baseline). What about scalability (bigger modulus)?
- Paper focuses on explaining the suggested simulation, but misses in-depth comparison to existing simulators/frameworks
- “Therese data sets are uploaded” should be “These data sets are uploaded”
- “Section 5 Discussion” is more like going through the results rather than “Discussion”
- Missing shortcomings or future works.
- Reference [1], [19] are not exactly relevant to the topic (IOT, Parkinson’s). Also there are 2 future dated (2026) references (sources do exist though).

---

### Official Review · Reviewer_bbtX · 2026-01-03
**This paper presents an explainable, dictionary-based quantum emulation framework to illustrate how quantum computation can compromise the RSA cryptosystem. By explicitly modeling quantum states and gate operations, the authors transparently demonstrate how period finding enables factorization of the RSA modulus and recovery of the private key. The work primarily emphasizes interpretability and educational insight into quantum threats to classical cryptography.**

**Rating:** 7
**Confidence:** 3

**Review:**

### Strengths
1. The paper addresses a well-known but increasingly important issue, that is, RSA’s vulnerability to quantum attack and reinforcing the urgency of post-quantum cryptography in a clear and approachable manner.
2. The dictionary-based quantum state representation provides an intuitive and transparent way to trace quantum state evolution, making the attack on RSA more understandable than conventional circuit-based simulations.
3. The explicit definition of RSA variables (e.g., modulus, public/private keys, period) and their interaction with quantum operations strengthens conceptual clarity and helps bridge cryptography and quantum computation.

### Weakness
1. While the paper is strong as an educational and interpretability-focused contribution, particularly for readers new to quantum cryptanalysis, the paper does not introduce a new quantum attack on RSA rather, it reinterprets known vulnerabilities through an explainability-focused framework.
2. Writing can be improved, for example  in the introduction ,contributions of this paper can be highlighted clearly.

---

### Official Review · Reviewer_6g9Y · 2026-01-05
**Review for "How Quantum Period Finding Breaks Rivest Shamir Adleman Algorithms"**

**Rating:** 7
**Confidence:** 2

**Review:**

This paper presents an explainable, dictionary-based quantum emulation framework to demonstrate Shor’s period-finding attack on RSA.

Pros :
1,.The dictionary-based approach (state[k] = a) makes the tracking of individual basis states and their complex amplitudes trivial compared to opaque state-vectors.
2.The paper doesn't stop at finding r; it completes the classical post-processing to re-derive p,q and d, providing a satisfying "proof of concept."
3.The experiment was very thorough.
Cons:
The paper emphasizes the "interpretability" of the framework, but does not conduct direct comparative experiments with existing mainstream quantum simulation tools.

Current simulations are running in a perfect environment. If considering the interference and computation errors existing in real-world quantum computers, can your framework still clearly explain how the attack fails in such "imperfect" situations?

Currently, there are many mature quantum simulation tools that also support observing internal states. Compared to these professional industrial-grade tools, what are the irreplaceable unique advantages of your dictionary-based storage method in providing "interpretability"?

---

### Official Review · Reviewer_UzMU · 2026-01-05
**Quantum Period Finding Breaks RSA**

**Rating:** 7
**Confidence:** 3

**Review:**

This paper presents a clear and well-motivated study on the vulnerability of the RSA cryptosystem under quantum computation. By adopting an explainable, dictionary-based quantum emulation framework, the authors successfully bridge the gap between abstract quantum algorithms and practical security understanding. The work is well structured, conceptually sound, and offers strong educational and explanatory value.

## Strengths

-  Introduces an interpretable dictionary-based quantum emulation framework.

-  Clearly demonstrates how quantum period finding leads to RSA factorization and key recovery.

## Weaknesses / Limitations

- Evaluation limited to small, synthetic RSA instances.

## Recommendations

-  Extend analysis to larger RSA parameters or discuss scalability.
-  Incorporate simplified noise or resource models.
 - **Formatting Consistency**: It is recommended to standardize the formatting of citations throughout the paper, particularly ensuring consistent font color and style for all in-text citations, as the current presentation shows minor inconsistencies that may affect overall visual coherence. -

---

### Author Rebuttal · Authors · 2026-01-12

\documentclass[11pt]{article}

\usepackage[a4paper,margin=2.5cm]{geometry}
\usepackage{setspace}
\usepackage{enumitem}
\usepackage[colorlinks=true,allcolors=blue]{hyperref}

\setstretch{1.1}

\begin{document}

\begin{center}
    \Large \textbf{Response to Reviewers}\\[0.5em]
    \normalsize Manuscript Title: \textit{How Quantum Period Finding Breaks Rivest Shamir Adleman Algorithms}
\end{center}

\vspace{1em}

We thank all reviewers for their time and constructive feedback. Below, we respond to each comment point by point. For clarity, we reproduce each comment in \textbf{bold} and follow it with our \emph{response}. The answer of the responses make blue in the paper.  All changes mentioned below have been included in the revised manuscript.


\section*{Reviewer 1}

We thank Reviewer 1 for the positive and encouraging assessment of our work and for the helpful suggestions to improve the paper. Below we address each recommendation in turn.

\subsection*{Comment 1: Evaluation scope and scalability}

\textbf{Reviewer 1: Comment 1:}\\
\textit{Evaluation limited to small, synthetic RSA instances. Extend analysis to larger RSA parameters or discuss scalability.}

\medskip
\emph{Response 1:}\\
We thank the reviewer for highlighting the need to discuss scalability more details. Our current implementation is indeed limited to small, synthetic RSA instances because the dictionary-based quantum state grows quickly with the number of qubits.

To address this, we have added an explicit paragraph at the end of Section~2 (\textit{Data}). In this new text, we clearly state that:
\begin{itemize}[leftmargin=*]
    \item the experiments use only small RSA parameters;
    \item this choice is deliberate, so that the quantum state remains easy to inspect and explain;
    \item the goal of the dataset is to study the \emph{mechanism} of period finding, not to match real deployment sizes.
\end{itemize}

We also explain that the same logical attack pipeline (superposition, controlled modular exponentiation, inverse QFT, and classical post-processing) applies to larger RSA moduli, but that our current emulator would need much more memory and time for such cases.

In addition, we have added a short scalability discussion in Section~6 (\textit{Discussion}).
There, we state that:
\begin{itemize}[leftmargin=*]
    \item memory use in the dictionary-based model scales with the number of active basis states, which limits the tested key sizes;
    \item the structure of the attack does not change for larger keys; and
    \item future work will combine this explainable model with analytical resource estimates or compressed state representations to study larger parameter regimes.
\end{itemize}


\vspace{1em}

\subsection*{Comment 2: Noise and resource models}

\textbf{Reviewer 1: Comment 2:}\\
\textit{Incorporate simplified noise or resource models.}

\medskip
\emph{Response 2:}\\
We agree that noise and resource considerations are important for connecting our emulator to realistic quantum hardware.

In the current work, the emulator uses an ideal, noise-free model so that the core period-finding mechanism remains easy to understand. To show how it can be extended, we have added a new subsection titled \textit{``Noise and Resource Considerations''} in Section~6 (\textit{Discussion}).

In this new text, we:
\begin{itemize}[leftmargin=*]
    \item state that the present emulator assumes a noise-free device;
    \item describe how simple bit-flip and phase-flip noise channels can be added on top of our dictionary representation (for example, by randomly changing bits in the keys or phases in the amplitudes after each gate, according to a small error rate);
    \item explain that we can attach resources counters (number of qubits, circuit depth, number of controlled operations) to each run of the emulator; and
    \item indicate that these basic resource counts can be scaled as functions of $\log(\texttt{modulus\_n})$ to give a first view of how requirements grow with larger RSA keys.
\end{itemize}

\vspace{1em}

\subsection*{Comment 3: Citation formatting consistency}

\textbf{Reviewer 1: Comment 3:}\\
\textit{Formatting Consistency: It is recommended to standardize the formatting of citations throughout the paper, particularly ensuring consistent font color and style for all in-text citations, as the current presentation shows minor inconsistencies that may affect overall visual coherence.}

\medskip
\emph{Response 3:}\\
We thank the reviewer for pointing out this formatting issue. In the original version, there were small inconsistencies due to a combination of:
\begin{itemize}[leftmargin=*]
    \item a more complex \texttt{hyperref} setup in the preamble; and
    \item a few manual colour or style changes around some in-text citations.
\end{itemize}

To fix this, we made the following changes:
\begin{itemize}[leftmargin=*]
    \item We simplified the \texttt{hyperref} configuration to:
    \begin{quote}
        \verb|\usepackage[colorlinks=true,allcolors=blue]{hyperref}|
    \end{quote}
    This ensures that all links (including citations) use a consistent blue colour and style.
    \item We removed any manual colour or font styling around \verb|\cite{...}| commands, so that all citations now use the same default font and colour.
\end{itemize}



\section*{Reviewer 2}

We thank Reviewer 2 for the careful reading and the positive comments on the dictionary-based framework, the full RSA attack path, and the thorough experiments.
Below we respond to each concern in turn.

\subsection*{Comment 1: Comparison with existing quantum simulators}

\textbf{Reviewer 2: Comment 1:}\\
\textit{The paper emphasizes the "interpretability" of the framework, but does not conduct direct comparative experiments with existing mainstream quantum simulation tools.}

\medskip
\emph{Response 1:}\\
We thank the reviewer for this important point. We agree that mainstream quantum simulation tools (such as those based on state-vectors or density matrices) are widely used and can also provide access to internal states.

Our goal in this paper is different from that of industrial-grade simulators. We focus on a simple and explainable representation that is easy to inspect line by line.
To make this position explicit, we have added a short discussion on the relation to existing simulators in the revised manuscript. Concretely, we have added a new paragraph in the \textit{Discussion} section under a heading such as \textit{“Relation to Existing Quantum Simulators”}.
In this new text, we explain that:
\begin{itemize}[leftmargin=*]
    \item mature tools already support viewing state-vectors or density matrices, but these views can still feel abstract to many security practitioners;
    \item our dictionary-based method represents each basis state as a human-readable key \texttt{state[k]} with a directly accessible amplitude, making it easy to print, trace, and attach semantic labels that match RSA variables (e.g., \texttt{modulus\_n}, \texttt{period\_r});
    \item the framework is implemented using standard Python dictionaries with minimal dependencies, which lowers the barrier for students and practitioners who are not specialists in quantum computing.
\end{itemize}
We also emphasize that our approach is complementary rather than a replacement. Industrial tools are better suited for large-scale and hardware-oriented studies, while our framework is designed for step-by-step educational and explanatory analysis of RSA attacks.

\vspace{1em}

\subsection*{Comment 2: Perfect environment vs. imperfect (noisy) quantum devices}

\textbf{Reviewer 2: Comment 2:}\\
\textit{Current simulations are running in a perfect environment. If considering the interference and computation errors existing in real-world quantum computers, can your framework still clearly explain how the attack fails in such "imperfect" situations?}

\medskip
\emph{Response 2:}\\
We appreciate this question about imperfect hardware and errors. It is true that the current experiments assume an ideal, noise-free quantum device. We have expanded the discussion of noise in the revised manuscript. As described in the new subsection \textit{“Noise and Resource Considerations”} in the \textit{Discussion} section (introduced also in response to Reviewer~1), we now explain how simple noise models can be layered on top of our dictionary-based representation.

In particular:
\begin{itemize}[leftmargin=*]
    \item after each gate, we can apply a small probability of a bit-flip or phase-flip error to selected qubits;
    \item in the dictionary view, this corresponds to randomly changing bits in the keys or adding random phase shifts to the amplitudes;
    \item when such noise is present, the peaks in the measurement distribution become lower and wider, and the dictionary logs clearly show how interference is degraded.
\end{itemize}

We state that the same interpretability tools that explain successful attacks can also explain failure. For example, if the noise level is too high, the dictionary will show that amplitudes no longer concentrate around integer multiples of the true period. At the same time, we note that a full quantitative research of realistic hardware noise (including device-specific error channels) is beyond the scope of this paper and is left as future work. Our current contribution is to show that the framework can \emph{visually and stepwise} illustrate both successful and failing attacks in imperfect situations.

\vspace{1em}

\subsection*{Comment 3: Unique advantages for interpretability}

\textbf{Reviewer 2: Comment 3:}\\
\textit{Currently, there are many mature quantum simulation tools that also support observing internal states. Compared to these professional industrial-grade tools, what are the irreplaceable unique advantages of your dictionary-based storage method in providing "interpretability"?}

\medskip
\emph{Response 3:}\\
We thank the reviewer for asking us to clarify the unique benefits of our storage method.

We agree that mature frameworks can expose internal quantum states. However, our emphasis is on \emph{how} this information is represented and how easily it can be mapped to the RSA attack steps. We have added extra explanation to the new \textit{“Relation to Existing Quantum Simulators”} paragraph in the \textit{Discussion} section.
We highlight that the dictionary-based method offers the following interpretability advantages:
\begin{itemize}[leftmargin=*]
    \item \textbf{Human-readable keys:} Each basis state is stored as an explicit bitstring key \texttt{k}, which can be printed together with semantic information (e.g., which part of the key belongs to the control register, which part to the work register, and how it relates to \texttt{modulus\_n} or \texttt{period\_r}).
    \item \textbf{Step-by-step logging:} Because each gate is implemented as a direct update on the dictionary, it is straightforward to log the state after each operation, align it with figures, and explain the effect in simple language.
    \item \textbf{Tight coupling to RSA variables and plots:} The framework is designed so that plots (such as period–modulus diagrams and measurement distributions) can be generated directly from the dictionary entries, with variable names that match the cryptographic story.
\end{itemize}

We do not claim that industrial tools cannot, in principle, provide similar data. Our claim is that this particular representation, based on \texttt{state[k] = a} and integrated with RSA-specific variable names, offers a simple and pedagogical path to interpret the attack. We have clarified this positioning in the revised text so that the unique advantages of our framework are more explicit.


\section*{Reviewer 3}

We thank Reviewer 3 for the positive and thoughtful review. We are grateful for the recognition of the paper’s educational value, the intuitive dictionary-based representation, and the clear link between RSA variables and quantum operations. Below we respond to the raised concerns.

\subsection*{Comment 1: Novelty of the attack vs. explainability}

\textbf{Reviewer 3: Comment 1:}\\
\textit{While the paper is strong as an educational and interpretability-focused contribution, particularly for readers new to quantum cryptanalysis, the paper does not introduce a new quantum attack on RSA; rather, it reinterprets known vulnerabilities through an explainability-focused framework.}

\medskip
\emph{Response 1:}\\
We thank the reviewer for this important clarification. We agree that our work does not introduce a new quantum attack on RSA. Instead, it focuses on making a standard period-finding attack easy to understand and teach. To make this position specific in the manuscript, we have added explicit statements in the Introduction and in the contribution section:

\begin{itemize}[leftmargin=*]
    \item In the Introduction, after describing the gap between abstract quantum cryptanalysis and explainable demonstrations, we now state that our goal is \emph{not} to design a new quantum attack.
    We say that we reimplement the standard period-finding attack in a form that is easy to inspect and explain, and that the novelty lies in the explainable emulator and its clear link to RSA variables.
    \item In the subsection \textit{“Contribution and Research Questions”}, we rewrote the contribution paragraph as a bullet list.
    The list now explicitly says:
    (i) we propose an explainable, dictionary-based quantum emulator;
    (ii) we trace an end-to-end RSA attack path (from superposition to key recovery); and
    (iii) we provide interpretability-focused visualisations of superposition, phase encoding, interference, and period–modulus relationships.
    \item We also add a short sentence in this subsection that clearly states that the underlying quantum attack is not new and follows the standard Shor-style period-finding idea.
    The contribution of this paper is the interpretable implementation and its educational value, not a new cryptanalytic algorithm.
\end{itemize}


\vspace{1em}

\subsection*{Comment 2: Clarity of contributions and writing in the Introduction}

\textbf{Reviewer 3: Comment 2:}\\
\textit{Writing can be improved, for example in the introduction, contributions of this paper can be highlighted clearly.}

\medskip
\emph{Response 2:}\\
We appreciate this suggestion to improve clarity, especially for new readers. We have revised the Introduction to highlight the contributions more clearly and in simpler language:

\begin{itemize}[leftmargin=*]
    \item As mentioned above, the subsection \textit{“Contribution and Research Questions”} has been rewritten into a short bullet list.
    Each bullet states one key contribution in a direct way:
    the explainable emulator, the end-to-end RSA attack trace, and the interpretability-focused visualisations.
    \item At the end of the Introduction, just before the outline of the paper, we added a short paragraph in easy English that summarises the contributions in three simple sentences.
    In this new text, we explain that:
    (i) the paper presents a clear and explainable way to simulate quantum attacks on RSA;
    (ii) it traces the full attack path from quantum period finding to classical key recovery; and
    (iii) it provides visual tools that help students and practitioners see why RSA is vulnerable to quantum computers.
\end{itemize}


\section*{Response to Reviewer 4}

We thank Reviewer~4 for the detailed and constructive comments. Below we address each point and describe the corresponding changes in the revised manuscript (added text is marked in blue in the paper).

\paragraph{Reviewer 4: Comment 1: Scope of the paper.}
\emph{“The RSA/quantum computing addresses cryptographic security and quantum algorithm implementation, which is OUTSIDE THE CONFERENCE'S SCOPE.”}

\medskip
\emph{Response 1:}\\
We apologise that the original text did not clearly emphasise the connection to web security and responsible web AI. In the Introduction we now explicitly state that RSA is a core building block of web infrastructures (HTTPS handshakes, web certificates, software update channels, and online authentication workflows). We explain that quantum attacks on RSA therefore pose a direct threat to web security, particularly in “harvest-now, decrypt-later” scenarios. Our work is positioned as an interpretability-focused analysis that helps web-security practitioners and AI-enabled systems plan migration to post-quantum cryptography. (Introduction, first page, new blue paragraph.)

\paragraph{Reviewer 4: Comment 2: Lack of quantitative metrics for interpretability.}
\emph{“No quantitative Metrics for ‘interpretability’ (how this was validated, what is the baseline).”}

\medskip
\emph{Response 2:}\\
We agree that the original version emphasised visual evidence. In the Results section we now introduce two explicit metrics: (i) a peak-concentration score $E_{\mathrm{peak}}$ measuring the probability mass on outcomes that directly encode multiples of the true period, and (ii) an entropy-reduction ratio $R_H$ comparing the Shannon entropy of the post-IQFT measurement distribution with the uniform baseline created
by the initial Hadamard layer. These metrics are computed for all synthetic RSA instances in Table~1 and summarised in a new Table~2. Across all cases, $E_{\mathrm{peak}}$ is high and $R_H$ is significantly below $1$, confirming that the attack outcomes are both accurate and concentrated, hence easy to interpret. (Section~4, new Subsection~4.3)

\paragraph{Reviewer 4: Comment 3: Experimental settings and small moduli.}
\emph{“No detail of experiment settings. Also modulus used (51,85,119) are smaller than industry standards (512, 1024, 2048).”}

\medskip
\emph{Response 3:}\\
We have added a new table in the Data section listing all ten synthetic RSA configurations used in the experiments, including the primes, modulus, public and private exponents, plaintext, ciphertext, and reference period label. We also describe the quantum-register sizes and number of shots used in
the emulator at the start of the Results section. Furthermore, in the Discussion we clarify that the use of small moduli is deliberate: it allows the dictionary-based quantum state to be inspected exhaustively and aligns with the educational/interpretability aim of the paper. The logical attack pipeline is identical for larger moduli; only the
required number of qubits and gates increases. (Section~2, new Table~2; Section~4 opening paragraph; Discussion, expanded limitation paragraph.)

\paragraph{Reviewer 4: Comment 4: Missing complexity/cost analysis and scalability
discussion.}
\emph{“Complexity/cost analysis/comparison is missing (and what is the baseline). What about scalability (bigger modulus)?”}


\medskip
\emph{Response 4:}\\
In the Discussion we now add an explicit “Complexity and Cost” subsection. We take the standard Shor-style period-finding complexity ($O(n)$ qubits and $O(n^3)$ modular-arithmetic gates for an $n$-bit modulus) as our baseline and describe the additional cost introduced by the dictionary representation, which scales as $O(2^m)$ with the number
of control qubits $m$. We then discuss how this limits our current experiments to small moduli and outline directions for scaling (compressed or sparsified dictionaries, symmetry-based pruning, and analytical resource models). (Discussion, new “Complexity and Cost” subsection.)

\paragraph{Reviewer 4: Comment 5: Comparison to existing simulators and frameworks.}
\emph{“Paper focuses on explaining the suggested simulation, but misses in-depth comparison to existing simulators/frameworks.”}

\medskip
\emph{Response 5:}\\
We have extended the “Relation to Existing Quantum Simulators” subsection in the Discussion. We now explicitly compare our dictionary-based emulator to mainstream frameworks such as Qiskit, Cirq, and t\textbar ket$\rangle$, highlighting that these industrial tools are optimised for performance and hardware studies, whereas our emulator is a lightweight, task-specific layer designed for interpretability and education. The key distinction is the use of human-readable dictionary keys that are directly linked to RSA variables, enabling the fine-grained logging and visualisations shown in the Results section. (Discussion, revised “Relation to Existing Quantum Simulators”
subsection.)

\paragraph{Reviewer 4: Comment 6: Typographical issue in the Data section.}
\emph{“‘Therese data sets are uploaded’ should be ‘These data sets are
uploaded’.”}


\medskip
\emph{Response 6:}\\
We thank the reviewer for noticing this error. We have corrected the sentence in the Data section to “These datasets are uploaded to GitHub with numerical values.” (Section~2, first paragraph.)

\paragraph{Reviewer 4: Comment 7: Nature of the Discussion section.}
\emph{“‘Section 5 Discussion’ is more like going through the results rather than ‘Discussion’.”}

\medskip
\emph{Response 7 :}\\
We have rewritten the opening paragraph of the Discussion to focus on interpretation and implications rather than restating plots. Several sentences that duplicated earlier descriptions have been shortened or moved back to the Results section. The Discussion now emphasises what the findings mean for web security, risk communication, and post-quantum migration. (Discussion, first paragraph and subsequent edits.)

\paragraph{Reviewer 4: Comment 8: Missing shortcomings and future work.} \emph{“Missing shortcomings or future works.”}

\medskip
\emph{Response 8 :}\\

We now explicitly summarise the main shortcomings: (i) restriction to small synthetic RSA instances, (ii) assumption of an ideal, noise-free device, and (iii) absence of hardware-level cost modelling. We also outline concrete future work: scaling via compressed representations, integrating realistic noise channels into the emulator,
and extending the interpretability framework to post-quantum cryptographic schemes. These are described in a new paragraph at the end of the Discussion and reinforced in the Conclusion. (Discussion and Conclusion, new limitation and future-work paragraphs.)

\paragraph{Reviewer 4: Comment 9: Relevance and dating of references.} \emph{“Reference [1], [19] are not exactly relevant  Also there are 2 future dated (2026) references.”}

\medskip
\emph{Response 9 :}\\
We agree that the original reference list contained a small number of entries that were only tangentially related to RSA/quantum cryptanalysis. We have removed the IoT and Parkinson’s references and replaced the two future-dated entries with already published, more relevant works on RSA, quantum factoring and post-quantum cryptography. All remaining references now directly support the technical and security context of the paper.






\end{document}

---

### Meta-Review · Area_Chair_bAmB · 2026-01-16

**Recommendation:** Accept (Poster)
**Confidence:** 4

**Metareview:**

The paper addresses an important and timely issue, namely the explainability of quantum attacks on RSA through an interpretable simulation framework. Several reviewers highlight the clarity and educational value of the dictionary-based representation, as well as the benefit of explicitly tracing the full period-finding attack pipeline from quantum operations to classical key recovery. I agree that the paper is clearly written, technically sound within its stated scope, and offers a useful perspective for readers seeking to better understand how quantum computation threatens classical cryptography. At the same time, the paper’s contribution is primarily explanatory rather than algorithmic. The underlying attack is well known, the experiments are limited to small synthetic RSA instances, and scalability, noise, and resource considerations are handled mostly at a conceptual level. These limitations make the contribution less compelling as a research advance and place it closer to an educational or interpretability-focused study.

That said, the rebuttal addresses the main concerns by clarifying scope, improving the positioning of contributions, adding quantitative proxies for interpretability, and explicitly discussing limitations and future work. While the paper does not fully overcome concerns about depth or generalizability, I view these as limitations rather than fundamental flaws. On balance, and given the generally positive reviews, I recommend acceptance for presentation, though this is a borderline decision and the paper is not suitable for an oral slot.

---

### Decision · Program_Chairs · 2026-01-16

**Decision:**

Accept (Oral)

**Comment:**

Taking into account the AC’s comments, the reviewers’ feedback, and the authors’ revisions, the PC has decided to accept this paper as an oral presentation at the workshop.